# UVC Up-Conversion and Vis-NIR Luminescence Examined in SrO-CaO-MgO-SiO_2_ Glasses Doped with Pr^3+^

**DOI:** 10.3390/ma17081771

**Published:** 2024-04-12

**Authors:** Olha Bezkrovna, Radosław Lisiecki, Bogusław Macalik, Przemysław Jacek Dereń

**Affiliations:** 1Institute of Low Temperature and Structure Research, Polish Academy of Sciences, ul. Okólna 2, 50-422 Wrocław, Poland; o.bezkrovna@intibs.pl (O.B.); r.lisiecki@intibs.pl (R.L.); b.macalik@intibs.pl (B.M.); 2Institute for Single Crystals, NAS of Ukraine, Nauky Ave. 60, 61001 Kharkiv, Ukraine

**Keywords:** optical glasses, UVC up-conversion, Pr luminescence

## Abstract

The application of ultraviolet-C light in the field of surface treatment or photodynamic therapy is highly prospective. In this regard, the stable fluorescent silicate SrO-CaO-MgO-SiO_2_-Pr_2_O_3_ glasses able to effectively convert visible excitation on the ultraviolet praseodymium emission were fabricated and examined. An unusual wide-range visible-to-UVC up-conversion within 240–410 nm has been achieved in Pr^3+^-doped glasses, revealing their potential advantage in different sophisticated disinfection technologies. The integrated emission intensity was studied as a function of light excitation power to assess a mechanism attributed to UVC luminescence. Especially, it was revealed that the multicomponent silicate glass qualities and praseodymium ^3^P_J_ excited state peculiarities are favorable to obtaining useful broadband ultraviolet up-converted luminescence. The glass dispersion qualities were determined between 450–2300 nm. The impact of praseodymium concentration on Vis-NIR spectroscopic glass qualities was evaluated employing absorption spectra, emission spectra, and decay curves of luminescence associated with two involved praseodymium excited states. Especially, efficient interionic interactions can be inferred by investigating the decrease in ^1^D_2_ state experimental lifetime in the heavily doped samples. Examination of absorption spectra as a function of temperature implied that excitation at 445 nm should be quite effective up to T = 625 K. Contrary to this, temperature elevation gives rise to a moderate lowering of the visible praseodymium luminescence.

## 1. Introduction

Many diseases are transmitted by airborne droplets and contact surfaces on which viruses, streptococci, staphylococci, and other microbes settle. This problem becomes particularly relevant during pandemics, the latest being the COVID-19 coronavirus. There is a known method for treating surfaces using ultraviolet lamps; however, they are dangerous for the health of people and animals. The significance of creating stable materials that emit efficiently in the UV region of the spectrum near 250 nm is due to the need to treat surfaces without the use of ultraviolet lamps [1]. The creation of materials that absorb photons from ambient light or sunlight and are capable of converting them into photons in the bactericidal range (220–280 nm) and their application can lead to a constant reduction in the content of adsorbed microbes. Ceramic powder materials based on fluorides and silicates Lu_7_O_6_F_9_:Pr^3+^ and Y_2_SiO_5_:Pr^3+^ are described in [2], as well as UV-resistant phosphors such as Ca_2_Al_2_SiO_7_:Pr^3+^ [3], which can be used to create such self-sterilizing surfaces.

In [4], a detailed analysis of saturation effects occurred in various fields of nonlinear optics and considered the nonlinear optical properties of various optical materials with a fast nonlinear optical response, which can be promising candidates for photonic applications, such as optical communications, optical limiters, optical data storage, information processing, passive laser mode-locking, etc. In [5], several multiphoton active materials and major applications of multiphoton excitation were described, including pumped lasing to achieve tunable up-conversion of coherent light.

In addition, recently more and more attention has been paid to the creation of light-emitting diodes as efficient, energy-saving, and environmentally friendly semiconductor lighting. Fluorescent glasses are expected to replace phosphors for LEDs since they have advantages over conventional phosphors [6], such as uniform light emission, lower manufacturing costs, and better thermal stability.

Pr^3+^ ions are recognized as effective luminescent activators over a wide range of wavelengths. For UV radiation, Pr^3+^ ions have an advantage due to the corresponding scheme of energy levels and the characteristic 4f^2^→4f^1^5d^1^ radiation observed in a wide UV range, including UV-C (200–280 nm) [3]. Praseodymium ions are the most effective lanthanides for converting the visible spectrum into UVC due to their qualities, such as a relatively wide range of two-photon excitation, an energetically wide cluster of intermediate states, and the tendency of electrons at excitation in the 4f5d band to relaxation due to high-energy transitions favoring UV radiation [2]. Pr^3+^ ions can emit in the UV range when excited at the 4f5d level directly or through up-conversion (UC) mechanisms [7]. Visible-to-UV conversion of Pr^3+^ ions occurs due to the absorption of blue or violet light (430–490 nm) through ^3^H_4_→^3^P_J_ transitions, followed by the UC of the excitation energy in one Pr^3+^ ion through Excited State Absorption (ESA) or involving two Pr^3+^ ions Energy Transfer Up-conversion (ETU) [2,7]. In addition, Pr^3+^ ions as activators can produce greenish-blue or red emissions due to their excited energy levels ^3^P_0_ or ^1^D_2_ [8].

Visible UV up-conversion using Pr^3+^-doped materials are excellent candidates for self-disinfecting surfaces. For sterilization in the UV region, various durable matrices with introduced ions emitting in the UV region of the spectrum can be used. The role of such matrices can be performed by glass. The beneficial properties of various types of glass determine their use in various fields. The chemical composition of glass, the additive introduced, and the method of its preparation determine its use. Therefore, glass-based materials are used in fiber optics [9], as radioactive waste immobilizers [10], as bioactive materials [11], or glazes [12]. Due to all the above qualities, Pr^3+^ ions are preferable for introducing them into a stable matrix.

To introduce Pr^3+^ ions, we chose a glass-based SrO-CaO-MgO-SiO_2_ matrix. A few glasses of similar composition are known. The production of (40–62%)SiO_2_-(2–15%)ZrO_2_-(20–50%)SrO-(4–22.5%)MgO (wt%) glasses was studied in [13], and it was found that compositions with 50 wt.% silica base contributes to the production of the transparent glasses. The authors showed that the introduction of Mg^2+^ ions increased the chemical resistance of glasses when they were treated with alkalis. As shown in [14], alkaline earth metal ions (calcium, barium, strontium, and other ions) dissolve in the Si–O–Si glass phase and, accordingly, can be a substitute for potassium ions in the glass composition of SiO_2_-MgO-Al_2_O_3_-B_2_O_3_-MgF_2_-K_2_O-Li_2_O-AlPO_4_. The composition of bioactive glasses based on the system of 42%SiO_2_-34%CaO-6%P_2_O_5_-18-15.5%SrO-(0-2.5%)Al_2_O_3_ (in mol%) is described in [11]. The authors of this work indicate that Sr^2+^ and Ca^2+^ ions have similar ionic radii (0.94 Å for Ca^2+^ and 1.16 Å for Sr^2+^), and therefore SrO oxide can be replaced by CaO in the glass composition. In [15], xSrO-(45.55-x)CaO-29.44SiO_2_-10.28P_2_O_5_-14.73MgO (x = 0–5 mol %) bioactive compositions were synthesized through the sol-gel route for the growth of hydroxyapatite on the surface of the materials as bioimplants. Examined in [16], 50SiO_2_-10Al_2_O_3_-2MgO-20CaO-15SrO-3BaO-(0.1–1.5)Pr_2_O_3_ (in mol %) glasses may be used for efficient visible fiber laser operating at 610 nm and 640 nm wavelengths.

Glasses doped with Pr^3^+ ions demonstrate the potential advantages attributed to broad bands in the wide spectral range and broad-band emission due to the disordered local environment of optically active ions. Among oxide glasses, silicate glasses doped with rare earth ions generally have the broadest absorption and emission bands [16], enabling significant tuning of the useful spectral range. In addition, silicate glasses are characterized by mechanical strength and chemical resistance.

In our work, we synthesized glasses based on alkaline earth metal oxides (Sr, Ca, and Mg) and SiO_2_ doped with different concentrations of Pr^3+^ ions, and consequently, their structural, physicochemical, and spectroscopic properties, including absorption, UVC up-conversion, and down-conversion, were examined.

## 2. Materials and Methods

Reagents SrCO_3_ (≥99.9%, Sigma-Aldrich, St. Louis, MO, USA), CaCO_3_ (99.95%, Thermo Scientific, Waltham, MA, USA), MgO (Reachem, Mississauga, ON, Canada), Pr_2_O_3_ (99.9%, Chem PUR, Piekary Śląskie, Poland), and nanopower of SiO_2_ (99.5%, 10–20 nm particle size (BET), Aldrich) were used for the synthesis of glasses. Glasses doped with praseodymium ions were prepared with a composition (in mol %) of 14.27 SrO-14.27 CaO-14.27 MgO-57.19 SiO_2_-Pr_2_O_3_ (x = 0.036, 0.071, 0.179 mol % Pr_2_O_3_) (this is equivalent to 0.5; 1.0; 2.5 at.% Pr^3+^, respectively). The choice of active Pr3+ ion concentrations in SCMS glasses is determined by a tradeoff between the praseodymium absorption qualities, especially at the excitation wavelength of 445 nm, and the ability to measure the up-conversion spectra of the material in the ultraviolet region as well as an emission in a wide spectral range. The ionic radii of the involved ions are the following: Mg^2+^ (0.66 Å), Ca^2+^ (0.99 Å) [17], (0.94 Å [11]), Pr^3+^ (1.126 Å) [18], and Sr^2+^ (1.260 Å [18]).

Samples were synthesized by the following procedure: First, the prepared components were transferred to a mortar and thoroughly ground. Next, the powder batch in the crucible was heated to 1450 °C for 2 h. The melt was poured into a brass mold. The glasses were characterized by the X-ray diffraction method (X′Pert PRO PANalytical diffractometer, Ashland, VA, USA, non-monochromatic CuKα radiation). The measurements were carried out in the range of 10–100° of 2θ with a scanning rate of 2θ min^−1^ for 30 min.

The absorption spectra of the glass samples were measured using a UV-Vis-NIR spectrophotometer (Cary 5000, Agilent, 5301 Stevens Creek Blvd, Santa Clara, CA, USA) in the spectral range of 190–2500 nm. The UV-VIS emission and excitation spectra were measured using an FLS1000 fluorescence spectrometer (Edinburgh Instrument, Livingston, UK) with a 450 W xenon lamp as an excitation source and a Hamamatsu 928 photomultiplier. Up-converted luminescence was measured using a 445 nm diode laser and a UG5 filter (Thorlabs, Newton, NJ, USA). To record emission spectra in the UVC range, a VUV McPherson spectrometer equipped with a water-cooled deuterium lamp and a Hamamatsu photomultiplier R955P (Hamamatsu, 430-0852 2-25-7 Ryoke, Nakaku, Japan) were employed as well. The Linkam system was used to study the effect of temperature on the spectral properties of the glasses. Decay curves of Pr^3+^ luminescence were acquired utilizing an experimental set-up consisting of a Surelite Continuum optical parametric oscillator pumped by the third harmonic of a Nd:YAG laser (Amplitude Laser Group, San Jose, CA, USA), a GDM 1000 (Carl Zeiss, Jena, Germany) double grating monochromator, an R3896 Hamamatsu photomultiplier (Hamamatsu, 430-0852 2-25-7 Ryoke, Nakaku, Japan), and a Tektronix MDO-40-54-B–3 Mixed Domain Oscilloscope (Tektronix Inc., Beaverton, OR, USA).

## 3. Results and Discussion

### 3.1. XRD Analysis

X-ray diffraction analysis was carried out to study the structure of as-melted glasses. In the diffraction pattern (Figure 1a) of the SCMS:Pr^3+^ glass samples, only two very weak and diffuse peaks with wide halos can be identified (in the range of 2θ = 23–35° and 39–47°), confirming the amorphous nature of the sample. This, in turn, makes their interpretation very difficult. Furthermore, the presence of nanocrystalline (several nm in size) inclusions of silicate phases, which may have characteristic peaks in the regions 2θ = 23–35° and 39–47°, cannot be completely excluded. The most pronounced peak around 2θ = 28.8° is most likely an amorphous halo, which is characteristic of amorphous or highly disordered SiO_2_-based structures.

An intense peak at 28.81°, as well as less intense peaks in the region 2θ = 25.99–26.12° and 28.80–29.18°, are characteristic of the SiO_2_ crystal structure (ICDS code 96-153-2514). The position of the halo with a maximum near 28.8° is also close to the positions of the peaks of the SiO_2_ structure (coesite, ICDS code 96-900-0805), however, high pressure is required for the formation of this structure. Silicates also have peaks in the region of 30–47°.

The structure of CaMgSi_2_O_6_ (ICSD code 30522) has peaks in the range of 26–31° and in the range of 40–44°. However, CaMgSi_2_O_6_ does not have the characteristic intense peak near 28.8°, which is characteristic of our glasses and the SiO_2_ structures.

The significant blurring of the peak (Figure 1a) with a maximum near 28.8° indicates that the formation of silicate structures (if they occur) is insignificant. Thus, we came to the conclusion that our materials are highly disordered structures based on SiO_2_.

### 3.2. Absorption Spectra, Band Gap, and Refractive Index

Figure 2a shows the absorption spectra of the SCMS glasses doped with Pr^3+^ (0.5, 1.0, and 2.5% Pr^3+^) in the spectral range from 380 nm to 2200 nm. All observed absorption bands are due to electronic transitions from the ^3^H_4_ ground state. The concentration dependence of the absorption coefficient of the ^3^H_4_→^3^P_0_ transition of Pr^3+^ ions is linear and is presented in the inset of Figure 2a. As shown in [16], aluminosilicate glasses containing oxides of strontium, calcium, and barium, have no absorption peaks in the wavelength range from 400 nm to 2400 nm. Therefore, all the well-defined peaks observed in Figure 2a are due to the presence of Pr^3+^ ions in the glass samples.

After doping the glasses with Pr^3+^ ions, as shown in Figure 2a,b, absorption peaks are located at 441 and 447 nm (^3^Н_4_→^3^Р_2_ transition), 463 (^3^Н_4_→^1^I_6_), 471 nm (^3^Н_4_→^3^Р_1_), 482 nm (^3^Н_4_→^3^Р_0_), 586 nm (^3^Н_4_→^1^D_2_), 980 nm (^3^Н_4_→^1^G_4_), 1397 nm (^3^Н_4_→^3^F_4_), 1508 nm (^3^Н_4_→^3^F_3_), 1904 nm (^3^H_4_→^3^F_2_) and near 2200 nm (^3^Н_4_→^3^H_6_). These energy transitions are shown in the energy level diagram of the SMCS:2.5%Pr^3+^ glass shown in Figure 3.

In this case, it is worth noticing that the absorption band maxima corresponding to the transitions from the ground state ^3^H_4_ to the excited states ^3^P_j_ were close to those recorded in [16]. On the other hand, the absorption transitions terminated on ^1^D_2_, ^3^F_4_, ^3^F_3,_ and ^3^H_6_ Pr^3+^ excited states are slightly shifted to the bands described in [16]. It is the result of the different compositions of the compared silicate glass materials. The relevant aspect of this paper is related to UVC up-converted luminescence, which is excited at 445 nm. Since the used semiconductor excitation source provides quite a high-power density to the glass sample, the impact of higher temperatures on the involved praseodymium absorption bands is worth investigating. In relation, the praseodymium absorption bands attributed to ^3^Н_4_→^3^Р_J_ and ^3^Н_4_→^1^D_2_ transitions for the SCMS:2.5% Pr^3+^ glass were examined as a function of temperature between 300 K and 675 K, i.e., from 26.85 °C to 401.85 °C. Our findings indicate that the examined absorption bands are rather ineffectively broadened at elevated temperatures. Furthermore, the effect of temperature on the ^3^Н_4_→^3^Р_2_ absorption band is moderate up to T = 675 K, and consequently, the excitation of up-converted praseodymium emission at 445 nm should be efficient even at high powers of the laser diode. The thermal-broadening effect and slight decrease in the absorption coefficient may be perceived for the ^3^Н_4_→^1^D_2_ absorption band at 586 nm as well.

The optical band gap of SCMS:Pr^3+^ glass was calculated employing Equation (1), as it was presented in [19,20,21]:(αhν)^2^ = А(h*ν* − E*g*),(1)
where A—is a constant, h—is Planck’s constant, and *ν*—is the optical frequency. The optical band gap of 4.41 eV was obtained for the SCMS:0.5%Pr^3+^ glass. The refractive index (n) was calculated using Fresnel equations [22]. The optical band gap calculated for the SCMS:0.5%Pr^3+^ glass is close to the values of 4.44–4.50 eV obtained in [22] for TiO_2_-SiO_2_ mixed thin films.

Figure 4 shows the measured refractive index (n) of SCMS:2.5% Pr^3+^ glass as a function of the incident wavelengths. The dispersive qualities of our silicate glass were examined at wide wavelengths ranging from near-infrared to UV. Concerning that, the value of the refractive index is lowered with the wavelength increasing to 1.56 at 2400 nm. For comparison, the comparable findings were described in [23] for the borosilicate glasses.

### 3.3. Up-Conversion Phenomena

We obtained up-convection luminescent radiation in the ultraviolet region in the SCMS:Pr^3+^ glass upon excitations with lower-frequency radiation. As shown in Figure 5, the up-converted luminescence spectrum of Pr^3+^-doped SCMS glass contains a broad band in the 230–330 UV spectral region. The maximum of the up-converted luminescence of the SCMS:Pr^3+^ glass under the 445 nm excitation is located at 275 nm.

To obtain more insight into the excitation mechanism of the praseodymium UV up-converted emission in the studied silicate glasses, the impact of the incident laser beam power on the integrated anti-Stokes emission was investigated. The result presented in Figure 6 shows the dependence of the up-converted ultraviolet emission on the 445 nm laser diode excitation power. The plot can very well be approximated by a straight line, and it shows that the excitation power is too low to induce the saturation effect. The slopes of the line indicate that the two-photon excitation process is responsible for up-converted emission in the SCMS:Pr^3+^ silicate glass.

The accomplishment of the ultraviolet anti-Stokes praseodymium emission in the studied silicate glasses was confirmed by employing an alternative experimental setup based on an FLS1000 spectrofluorometer.

In contrast to previously described spectra, the currently used detector allows for measuring the up-converted Pr^3+^ emission at longer wavelengths up to 410 nm. On the other hand, its sensitivity is significantly reduced at a higher-energy spectral range. For that, additional broad-band up-converted emission was observed within the 320–410 nm region, and these are presented in Figure 7. The intense emission band is peaked at 392 nm and the weaker UV band is centered at 360 nm. The origin of these bands is explained in the next chapter which describes the excitation and down-converted spectra.

It is recognized that praseodymium ions emit in the UV region of the spectrum in various hosts, including silicate ones. In fact, in [24], authors display that Pr^3+^ ions in LaBSiO_5_ powders reveal an intense luminescence band in the 220–280 nm region with a maximum near 231 nm. Xinshun Wang et al. documented up-converted Pr^3+^ luminescence in the UV-350 nm region in the Y_2_SiO_5_:Pr^3+^ crystal upon excitation by an infrared femtosecond laser at 800 nm [25]. The YSO:Pr host was investigated by E.L. Cates et al. [2], and as a result, the up-conversion spectrum was between 265–360 nm with a maximum of 278 nm under 447 nm excitation. Quite recently, for the other silicate crystal Y_2_Si_2_O_7_:Pr^3+^ [26], two ultraviolet broad bands ranging between ~250 nm to ~390 nm with maximum intensity at 278 nm and 308 nm were achieved under excitation at 445 nm. According to the author’s explanation, this resulting anti-Stokes UV praseodymium emission is a consequence of the sequential absorption of two blue photons, as confirmed by the measurements of excitation power dependence.

It Is worth noticing that values ‘f th’ energy gap Eg = 4.82 eV and 4.78 eV were reported for Y_2_SiO_5_ and Y_2_Si_2_O_7_ silicates, respectively [27]. These are slightly higher than Eg = 4.41 eV estimated for our SCMS glass. It can be concluded that during the effective excitation process at 445 nm, the ^3^P_J_ multiplets of Pr^3+^ ions are populated, after which, due to the UC process, the levels of the 4f^1^5d^1^ electronic configuration are fed, and as a result, UV anti-Stokes 4f^1^5d^1^→^3^H_J_ emission is observed.

### 3.4. Excitation and Down-Converted Luminescence Spectra

The optical properties of Pr^3+^ ions were studied in the UV, visible, and near-IR regions. The luminescence excitation spectra of SCMS glasses monitored at 605 nm and 730 nm are presented in Figure 8a,b. As can be seen in the 425–500 nm wavelength range, three peaks are noticeable at 441, 471, and 486 nm, caused by ^3^H_4_→^3^P_2_, ^3^H_4_→^3^P_1,_ and ^3^H_4_→^3^P_0_ transitions, respectively. These peaks are close to those presented in [6] for oxyfluoride silicate glasses doped with Pr^3+^ ions (~443, 468, and 481 nm—transitions from the ground state ^3^H_4_ to excited states ^3^P_2_, ^3^P_1,_ and ^3^P_0_, manifolds, respectively). It can be discerned that the spectral characteristic of the praseodymium excitation band is not affected by the concentration of Pr^3+^ ions in the material under study. In addition, as shown in Figure 8b, the luminescence excitation spectra of glasses contain a broad band in the UV region with a maximum at 273 nm related to ^3^H_J_-4f^1^5d^1^ inter-configurational transitions of praseodymium. This band is especially intense for the SCMS:2.5% Pr^3+^ glass with a high concentration of dopant.

An additional peak of the luminescence excitation band at 386 nm was observed for all SCMS glass samples. In [8], an additional peak with a maximum at 355 nm was also observed for the luminescence excitation spectra of La_2_Zr_2_O_7_:Pr^3+^ nanophosphors, and the authors associated the appearance of this peak with the intrinsic defect absorption or the virtual charge transfer of Pr^3+^ ions.

The appearance of a band with a maximum of 330 nm is characteristic of the luminescence excitation spectra of our glasses when detected at 385 nm (Figure 8c). When excited into this band at 335 nm, an intense luminescence band is observed (Figure 9c) with a maximum near 400 nm. A similar luminescence band was studied in [28] for the porous SiO_2_ matrix. The authors of this work showed that when the pure SiO_2_ matrix was excited at 335 nm, a broad luminescence band with a maximum at 380 nm (at room temperature) was observed and suggested that the nature of this band could be due to either isolated silanol groups or OH-related centers. However, during the synthesis of our glass, tetraethoxysilane was not used. We carried out the synthesis at high temperatures, so the more likely appearance of this band is associated with the presence of defects in the examined SCMS glasses.

In [29], it was stated that silicate glasses obtained in a reducing atmosphere (H_2_/He) can have a luminescence band with a maximum at 3.1 eV (400 nm) (strong) and 4.2 eV (295 nm) (very weak) with excitation at 5.17 eV (240 nm). The authors stated that such bands are also characteristic of studies in an oxidizing atmosphere. This band is identified as a Si-related center, which was assigned to Si(II), dissolved in the silica network.

It can be assumed that the presence of a band with a maximum at 400 nm in our glasses can also be due to the formation of chains of two-fold coordinated Si atoms with two oxygen neighbors; moreover, the presence of oxygen vacancies is also possible.

The luminescence spectra of the SCMS:Pr^3+^ glasses were measured in the visible and near-infrared regions. Samples of the SCMS glasses doped with Pr^3+^ were excited into absorption bands at 237 nm and 447 nm. The luminescence spectra of the doped glasses are presented in Figure 9a,b. The involved transitions are shown as the solid arrows in the energy level scheme of Pr^3+^ ions in the SCMS glasses (Figure 3).

When praseodymium luminescence is excited at 447 nm (Figure 9a), the emission bands appear with a maximum at 486 nm (^3^P_0_→^3^H_4_), 527 nm (^3^P_1_→^3^H_5_), and 552 nm (^3^P_0_→^3^H_5_). The most intense band with a maximum at 605 nm (^1^D_2_→^3^H_4_) contains weaker components near 631 nm (^3^P_0_→^3^H_6_) and at 645 nm (^3^P_0_→^3^F_2_). The remaining maxima can be discerned at 692 nm (^3^P_1_→^3^F_4_), 708 nm (^3^P_0_→^3^F_3_), 730 nm (^3^P_0_→^3^F_4_), 880 nm (^3^P_1_→^1^G_4_), and at 823 nm (^1^D_2_→^3^H_6_).

Figure 9b shows the luminescence spectrum excited at 237 nm, and consequently, the emission bands appear with maxima near 491 nm (^3^P_0_→^3^H_4_), 533 nm (^3^P_1_→^3^H_5_), 556 nm (^3^P_0_→^3^H_5_), the most intense 605 nm (^1^D_2_→^3^H_4_), with shoulder maxima at 631 nm (^3^P_0_→^3^H_6_), and 647 nm (^3^P_0_→^3^F_2_), as well as a band with a small maximum at 703 nm and 736 nm (^3^P_1_→^3^F_3_,_4_). Eventually, it can be perceived that in the visible luminescence spectra of the SCMS:Pr^3+^ glasses, red emission is prominent at ~605 nm (^1^D_2_→^3^H_4_ transition). The comparable branching ratio of praseodymium luminescence and the location of the examined peaks were documented in [30] for Pr^3+^-doped calcium aluminosilicate glasses. Moreover, the emission peaks of Pr^3+^ ions observed in our SCMS glasses coincide with the emission peaks reported in [6] for SiO_2_-Al_2_O_3_-CaO-CaF_2_-TiO_2_: Pr_2_O_3_ glasses excited at 443 nm.

It should be noted that when the glass samples are excited at 237, the maximum luminescence intensity of the ^1^D_2_-^3^H_4_ transition (605 nm) decreases for the SCMS:1% Pr^3+^ and SCMS:2.5% Pr^3+^ samples by a factor of 1.5–2 in relation to a glass with a low concentration of Pr^3+^ ions.

The decrease in praseodymium luminescence intensity related to the ^1^D_2_→^3^H_4_ transition near 605 nm with increasing dopant concentration may be due to cross-relaxation phenomena taking place between two neighboring optically active ions. As can be seen from Figure 9a, the ratio of peak intensities at 484 nm relative to the peak intensity at 605 nm is 0.18, 0.23, and 0.51, with an increase in Pr^3+^ ion concentration from 0.5 to 1.0, and 2.5%, respectively. Our findings indicated that visible emission of praseodymium in the SCMS glasses can be efficiently excited employing excitation bands at 441, 471, and 486 nm corresponding to ^3^H_4_→^3^P_2_, ^3^H_4_→^3^P_1_, and ^3^H_4_→^3^P_0_ transitions, and a higher-energy UV excitation is useful as well.

The luminescence of SCMS glass doped with 1% Pr^3+^ measured in the near-infrared wavelength range of 800–1700 nm with excitation at 447 nm is displayed in Figure 10. The NIR emission spectra consist of three bands that may be assigned to ^3^P_1_→^1^G_4_ (888 nm), ^1^D_2_→^3^F_4_ (1055 nm), and ^1^D_2_→^1^G_4_ (1496 nm) transitions. The band centered at 1055 nm is the most intense, and a transition terminated on the ^1^G_4_ level covers a wider 1300–1630 spectral range.

### 3.5. Impact of Temperature on Excitation and Luminescence Spectra

A study on the effect of 85 K (−188/15 °C)–715 K (441.85 °C) temperature on the luminescence excitation spectra of the SCMS:1% Pr^3+^ glass (^3^H_4_-^3^P_j_ transitions) has been prepared, and the resulting spectra are presented in Figure 11a. At higher temperatures, the higher-energy crystal field components of praseodymium ground state ^3^H_4_ are more effectively populated, consequently, the consecutive transitions to the involved sublevels of ^3^P_J_ multiplets take place. This effect, combined with the inherent thermal line shift and broadening, gives rise to the emission band extension, especially at longer wavelengths. In contrast to that, overall band intensity is reduced with temperature increasing. Particularly, at low temperatures, two high-energy components at 441 and 447 nm are more pronounced and become efficiently depressed at higher temperatures.

The praseodymium luminescence originating In ^3^P_J_ and ^1^D_2_ multiplets was measured as a function of temperature 85 K (−188/15 °C)–715 K (441.85 °C) for the SCMS glass doped with 1% Pr^3+^, and the adequate spectra are presented in Figure 11b. The most intense band at 605 nm is effectively broadened within shorter wavelengths, and the peak maximum is blue-shifted as well. It is a consequence of the population of higher-energy crystal field sublevels attributed to the involved praseodymium luminescent excited states.

We investigated the variation of luminescence intensity ratio (LIR) related to (^3^P_0_-^3^H_4_/^3^P_1_-^3^H_5_) praseodymium transitions as a function of 85–715 K temperature to evaluate the suitability of the studied materials for applications in optical sensing thermometry. The impact of temperature on the determined luminescence intensity ratio was used to estimate the corresponding absolute (S_A_) and relative (S_R_) thermal sensitivities for SCMS:1%Pr and SCMS:2.5%Pr glasses. The achieved results are depicted in Figure 12a–d. The luminescence intensity ratio was fitted according to the relation:LIR(T) = A + Bexp(∆E/(k_B_ T))(2)
where ∆E is the energy difference between the thermalized ^3^P_0,1_ levels, k_B_ is the Boltzmann constant, T is the temperature expressed in absolute scale [K] and A and B are constants.

Luminescence intensity ratios LIR (527/486) increase exponentially with temperature elevation, and the maximum absolute temperature sensitivity is reached for T = 223 K and T = 230 K, and amount to 8.71 × 10^−4^ K^−1^ and 9.83 × 10^−4^ K^−1^ for SCMS:1%Pr^3+^ and SCMS:2.5%Pr^3+^ respectively.

The highest values of relative sensitivity, S_R_ = 0.52% K^−1^ at T = 330 K and S_R_ = 0.48% K^−1^ at T = 322 K, were found for SCMS:1%Pr^3+^ and SCMS:2.5%Pr^3+^ glasses, respectively. Concerning that, our glasses can be considered a potential luminescent optical temperature sensor, applying thermally coupled levels of praseodymium. Quite recently, a maximum relative sensitivity of 1.0% K^−1^ has been reported for Pr^3+/^Yb^3+^ co-doped fluoride phosphate glass [31]. This estimated sensitivity is higher in relation to SCMS:Pr^3+^ glass, regardless of our optical systems can be useful at extended temperature ranges up to 715 K.

### 3.6. Relaxation Dynamic of Pr^3+^ Excited States

Luminescence decay curves were measured for ^3^P_0_ and ^1^D_2_ excited states at 495 nm or 605 nm under excitation at 447 nm. As can be seen from Table 1 and Figure 13a, the experimental lifetime of the ^3^P_0_ excited state changes slightly with increasing dopant concentration in the studied glass. In fact, an increase in the concentration of Pr^3+^ ions from 0.5% to 2.5% leads to an insignificant decrease in average lifetime *τ_exp._* of ^3^P_0_ levels from 2.20 μs to 2.06 μs.

A different effect is recognized for ^1^D_2_ levels of Pr^3+^-doped SCMS (Table 1 and Figure 13b). An increase in the concentration of Pr^3+^ ions leads to a considerable decrease in *τ_exp._* (from 187 μs to 74 μs at a concentration of Pr^3+^ ions of 0.5 and 2.5%, respectively). Our findings indicate that the efficiency of Pr-Pr energy transfer can be found to be around 60% in SCMS glasses.

To examine the interaction between optically active ions in SCMS:Pr^3+^ glasses, the Inokuti–Hirayama model has been utilized [32]. When the effectiveness of interionic energy transfer is more significant in relation to the time evolution of praseodymium, luminescence intensity can be described as:(3)$$\Phi \left(t\right)=Aexp\left[-\left(\frac{t}{\tau_{0}}\right)-\alpha {\left(\frac{t}{\tau_{0}}\right)}^{3/S}\right]$$
where *A* is constant, *Φ*(*t*) denotes the emission intensity after pulse excitation, *S* = 6 for dipole-dipole interactions, *τ*_0_ defines the intrinsic decay probability of the donor involved excited state when the acceptor is absent; furthermore, α is the parameter expressed as:(4)$$\alpha =\left(\frac{4}{3\pi }\right)\Gamma \left(1-\frac{3}{S}\right)N_{a}R_{0}^{3}$$
where *R*_0_ is the critical ion-ion energy transfer distance, *Na* is the acceptor concentration Γ = 1.77 (for *S* = 6) is Euler’s function. The nonexponential decay curve recorded for ^1^D_2_ luminescence in heavily doped SCMS:2.5%Pr^3+^ glass was applied to estimate energy transfer quality. In relation to that, the α parameter was fitted to be 1.9 and the critical energy transfer distance *R*_0_ is equal to 8.9 Å for the studied glass host. Employing the relations C_da_ = *R*_0_^6^*τ*_0_^−1^ and W_da_ = C_da_*R*_0_^−6^ the energy transfer parameter corresponds to the value of 2.40 × 10^−39^ cm^6^s^−1^ and a donor-acceptor energy transfer rate is estimated to be 5 × 10^6^ s^−1^. The estimated critical distance R_0_ is higher than 5 Å, hence the applied IH model validates the contribution of multipolar Pr-Pr interactions in the material under study [33]. For the comparison, the energy transfer parameters C_da_ = 6.39 × 10^−43^ cm^8^s^−1^ and W_da_ = 13 × 10^6^ s^−1^ were estimated for Pr-doped multi-component silicate photonic films [34]. In relation to our glass, this praseodymium highly-doped silicate optical system is characterized by a significant D-A energy transfer rate and especially dipole-quadrupole mechanisms leading to significant quenching phenomena. Contrary, for higher Pr_2_O_3_ concentration in P_2_O_5_-Na_2_O-Al_2_O_3_-Gd_2_O_3_ glass [35], particularly the reliable fitting is documented for *S* = 6 and dipole-dipole interaction between optically active ions takes place.

## 4. Conclusions

In this work, we studied UVC up-conversion phenomena and optical properties of SrO-CaO-MgO-SiO_2_-Pr_2_O_3_ glass matrix (0.5–2.5% Pr^3+^) in UV-vis-near-IR regions. Effective UVC up-converted emission within a wide spectral region was observed when 445 nm excitation into praseodymium ^3^P_J_ multiplets took place. The dependence of the integrated anti-Stokes emission intensity on the laser diode power indicates that the two-photon excitation process is responsible for the population of the involved luminescent levels. The luminescence of Pr^3+^ in the studied glasses mainly corresponds to peaks in blue (486 nm), reddish-orange (605 nm), and near-infrared (1055 nm) regions. The effect of temperature on praseodymium luminescence was examined in SCMS glass up to T = 715 K. Accordingly, the luminescence intensity ratio between thermally coupled (^3^P_0_-^3^H_4_/^3^P_1_-^3^H_5_) praseodymium excited states was utilized to estimate maximal relative temperature sensitivity, S_R_ = 0.52% K^−1^ at T = 330 K, for SCMS:1%Pr^3+^ glass. The considerable nonradiative energy transfer between praseodymium ions occurs, especially in glasses containing higher concentrations of luminescent ions. Relaxation dynamic study of ^1^D_2_ luminescence made it possible to estimate critical energy transfer distance R_0_, which amounts to 8.9 Å. Eventually, these findings indicate that the synthesized glasses can be suitable as solid-state media for various photonic applications, including the development of UVC self-sterilizing surfaces or temperature optical sensors.

## Figures and Tables

**Figure 1 materials-17-01771-f001:**
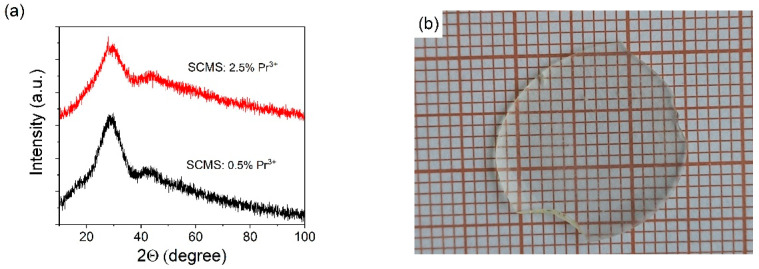
XRD of the SCMS:Pr^3+^ glasses (**a**), and the photo of the SCMS:1.0% Pr^3+^glass (**b**).

**Figure 2 materials-17-01771-f002:**
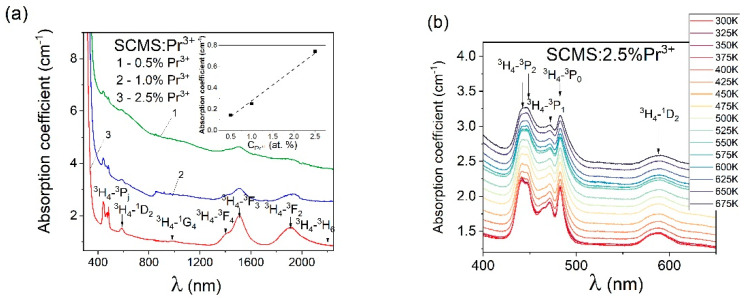
Absorption spectra of SCMS glasses doped with 0.5 (1), 1.0 (2), and 2.5% (3) Pr^3+^ ions (**a**) (inset shows the dependence of the absorption coefficient of ^3^H_4_→^3^P_0_ transition from Pr^3+^ concentration); effect of the temperature on SCMS:2.5% Pr^3+^ absorption bands (**b**).

**Figure 3 materials-17-01771-f003:**
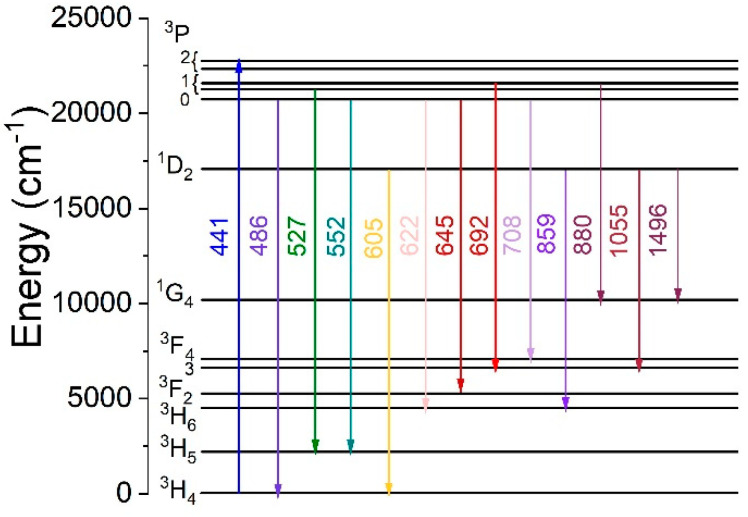
Energy levels scheme of Pr^3+^ ions in SCMS glasses.

**Figure 4 materials-17-01771-f004:**
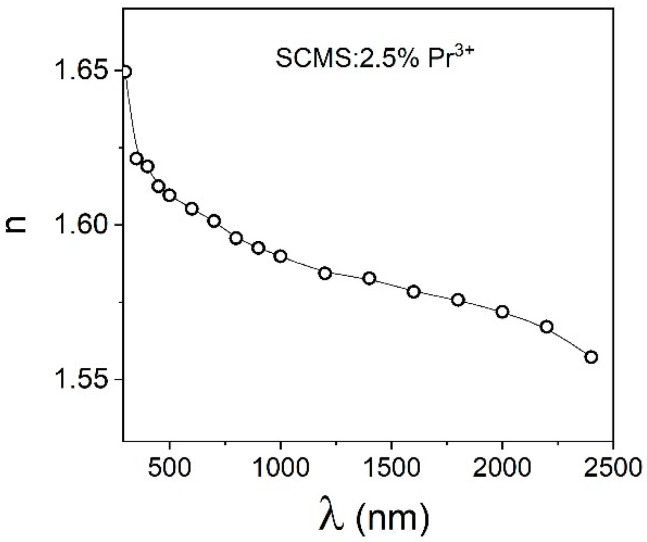
The dependence of the refractive indices of the SCMS:2.5% Pr^3+^ glass from the wavelengths.

**Figure 5 materials-17-01771-f005:**
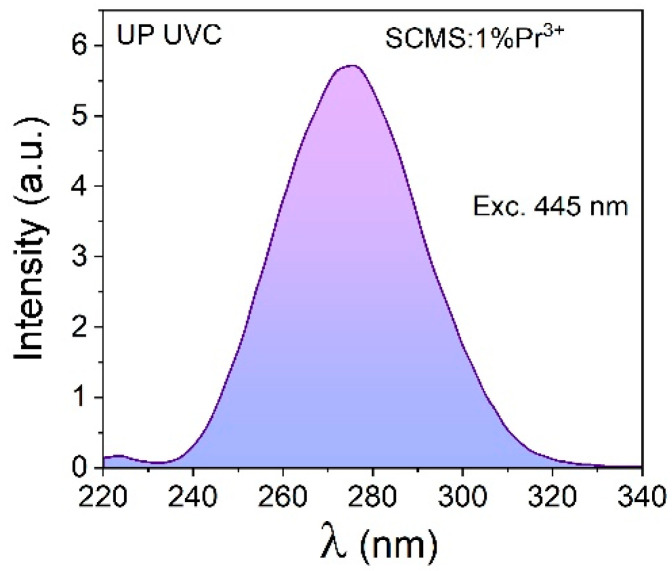
The up-converted UV luminescence of SCMS:1%Pr^3+^ glass excited at 445 nm.

**Figure 6 materials-17-01771-f006:**
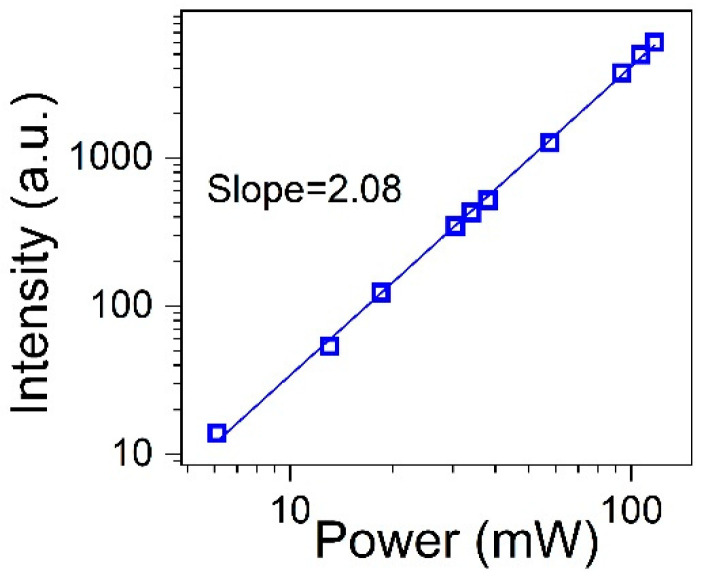
Integrated emission intensities vs. 445 nm CW excitation power for the SCMS:1%Pr^3+^ glass.

**Figure 7 materials-17-01771-f007:**
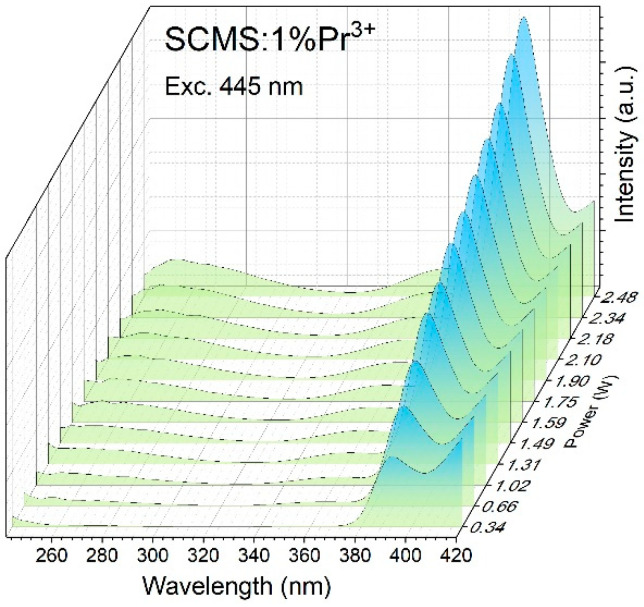
The anti-Stokes UV emission of Pr^3+^-doped SCMS glass excited at several excitation powers up to 2.48 W.

**Figure 8 materials-17-01771-f008:**
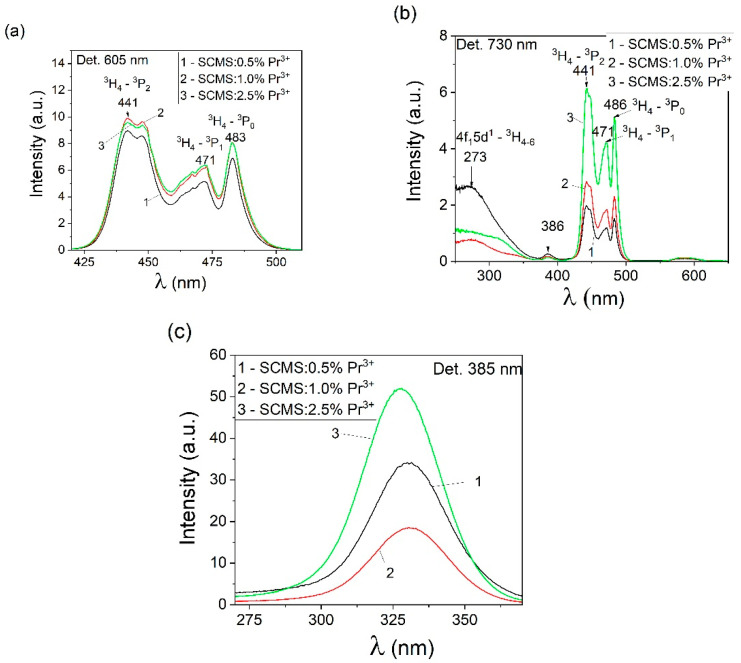
Luminescence excitation spectra of the SCMS:Pr^3+^ glasses with different concentrations of Pr^3+^ ions (1—0.5%, 2—1.0%, 3—2.5%) detected at 605 nm (**a**), 730 nm (**b**), and 385 nm (**c**).

**Figure 9 materials-17-01771-f009:**
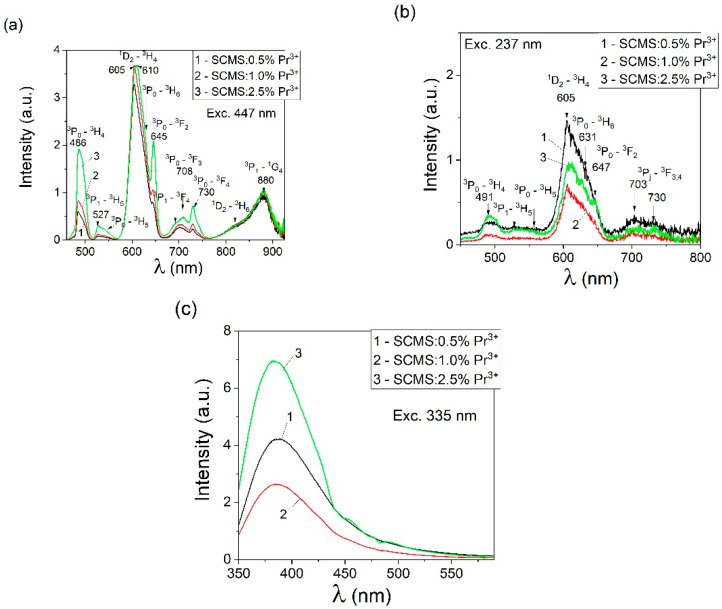
Luminescence spectra of the SCMS:Pr^3+^ glasses with the different concentrations of Pr^3+^ ions (0.5% (1), 1.0% (2), and 2.5% (3)) excited at 447 nm (**a**), 227 nm (**b**), and 335 nm (**c**).

**Figure 10 materials-17-01771-f010:**
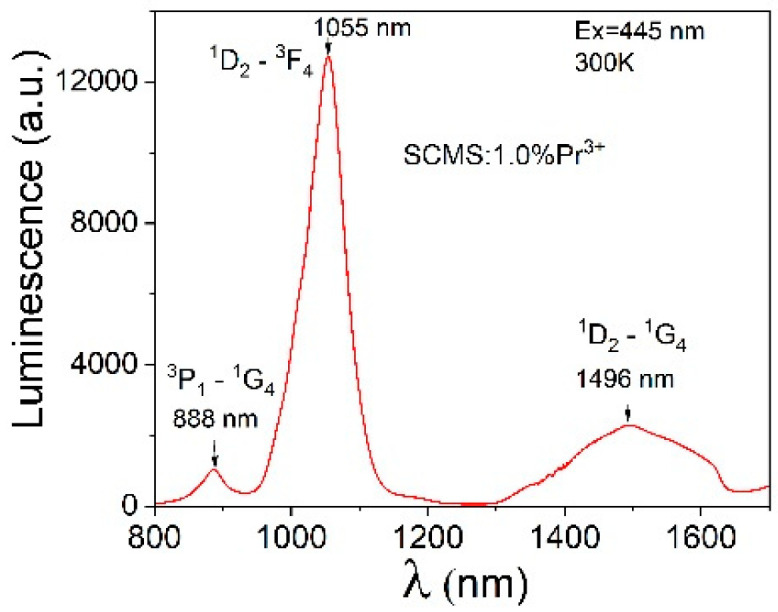
Near-infrared luminescence of the SCMS:1% Pr^3+^ glass.

**Figure 11 materials-17-01771-f011:**
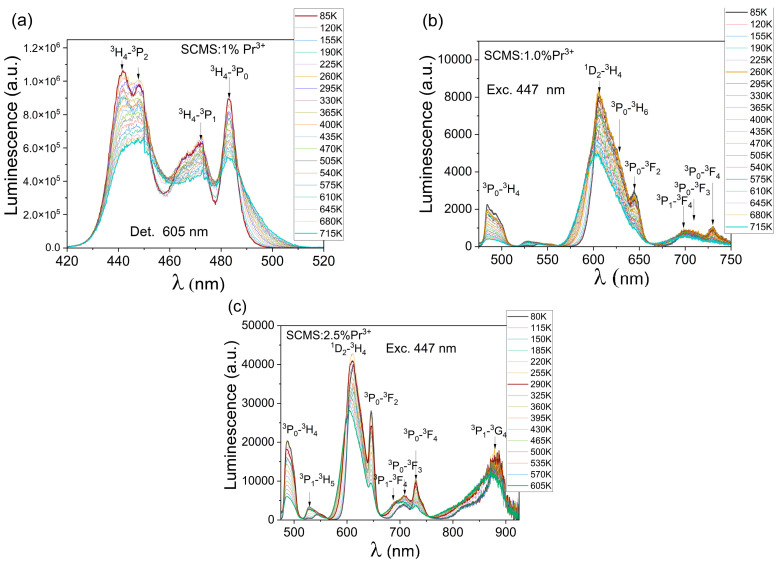
Luminescence excitation spectra of the SCMS:1% Pr^3+^ glass, λ_det_ = 605 nm (**a**) and luminescence spectra of the SCMS:1% Pr^3+^ (**b**) and SCMS:2.5% Pr^3+^ (**c**) glasses excited at 447 nm measured in a wide temperature range.

**Figure 12 materials-17-01771-f012:**
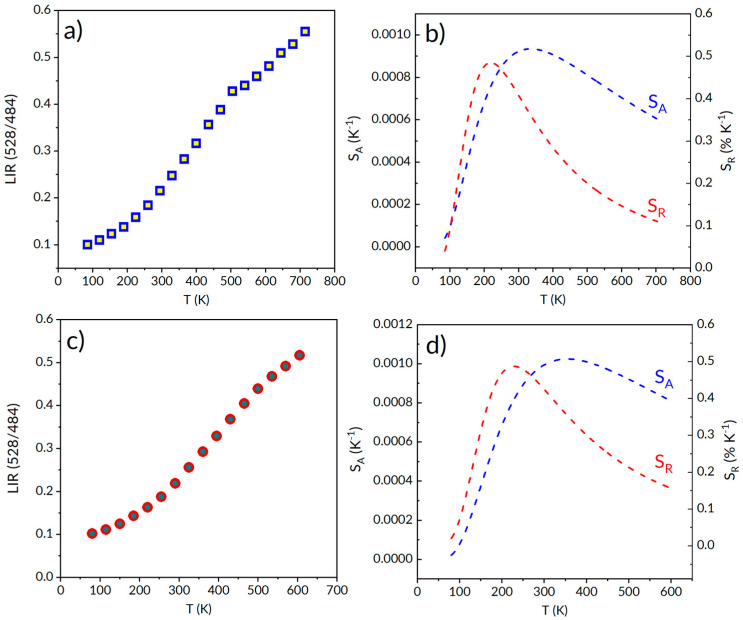
Luminescence intensity ratios correspond to praseodymium luminescence (**a**) of SMCS:1%Pr^3+^ (**a**,**b**) and SMCS:2.5%Pr^3+^ (**c**,**d**) glasses, as well as the corresponding absolute and relative thermal sensitivities estimated for LIR (527/486).

**Figure 13 materials-17-01771-f013:**
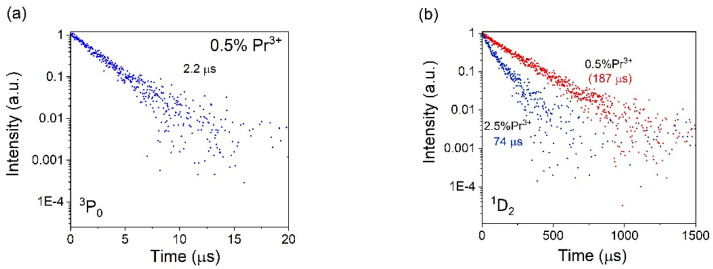
Photoluminescence decay profiles of ^3^P_0_ (**a**) and ^1^D_2_ (**b**) levels upon observation at 495 or 605 nm of Pr^3+^-doped SCMS glasses.

**Table 1 materials-17-01771-t001:** Experimental lifetimes of ^3^P_0_ and ^1^D_2_ levels of Pr^3+^ in SCMS glasses.

SCMS:Pr^3+^ Glasses	τ	
	^3^P_0_ [μs]	^1^D_2_ [μs]
0.5% Pr^3+^	2.20	187
1.0% Pr^3+^	2.12	145
2.5% Pr^3^	2.06	74

## Data Availability

The data presented in this study are openly available in Zenodo at https://doi.org/10.5281/zenodo.10817241.

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
