# Peer review of "UVC Up-Conversion and Vis-NIR Luminescence Examined in SrO-CaO-MgO-SiO2 Glasses Doped with Pr3+"

_materials, 2024, doi:10.3390/ma17081771_

Round 1

Reviewer 1 Report

Comments and Suggestions for Authors

The paper "UVC up-conversion and Vis-NIR luminescence examined in SrO-CaO-MgO-SiO2 glasses doped with Pr3+” present interesting findings.  The manuscript was written well. However, several points need to be addressed in this manuscript:

  1.     Page 3, line 129. It is stated that the samples do not show any diffraction peaks, but in figure 1 two broad peaks can be noted. The authors are urged to meticulously review the XRD analysis and rectify this omission.

2.      The figures 2, 8, 9 and 11 should be enlarged and presented at a higher resolution for better data clarity. Also, in figure 11c, the transitions must be specified.

3.      At page 7 to be replaced the words: assumed in ...is assumed in the next… (line 213); A capital letter and display in...in [22] Authors display that… (line 215), documented in ...spectrum was documented between.. (line 220) with more appropriate words. Suggestion: assumed with discussed, etc.

4.      At page 8, for a unified expression of the data in the paper with those in the literature, the band energies presented in lines 263-265 should also be expressed in nm.

5.      At page 12, line 359 (figure caption) to delete the I in (I527/I486)

6.       At page 13, the authors to give a brief explanation of the values obtained for the transfer energy and their correlation with the previously presented data.

Comments on the Quality of English Language

English is good with few grammatical and word meaning errors.

Author Response

Thank you very much for your comments. We addressed them all carefully, which is included below.

Remark 1. Page 3, line 129. It is stated that the samples do not show any diffraction peaks, but in figure 1 two broad peaks can be noted. The authors are urged to meticulously review the XRD analysis and rectify this omission.

Reply:

X-ray diffraction analysis was carried out to study the structure of as-melted glasses. In the diffraction pattern (Fig. 1a) of the SCMS:Pr3+ glass samples, only two very weak and diffuse peaks can be identified (in the range of 2θ = 23-35° and 39-47°), confirming the amorphous nature of the sample. This in turn makes their interpretation very difficult.

At the same time, the presence of nanocrystalline (several nm in size) inclusions of silicate phases, which may have characteristic peaks in the region of 2θ = 23-35° and 39-47°, cannot be completely excluded. The most pronounced peak around 2θ=28.8° is most likely an amorphous halo, which is characteristic for amorphous or highly disordered SiO2-based structures. SiO2 (space group C12/c1(15), crystal system – monoclinic; ICSD code 96-153-2514) can be formed, having peaks in the range of 2θ = 25.99-26.12°, an intense peak at 28.81° and less intense in the region of 28.80-29.18°. The position of the halo with a maximum near 28.8° is also close to the positions of the peaks of the SiO2 structure (coesite, ICDS code 96-900-0805), however, pressure is required for the formation of this structure. We did not apply an additional pressure, therefore, the formation of such a structure is difficult. Silicates also have peaks in the region of 30-47°. A study of the structure of CaMgSi2O6 (ICSD code 30522) showed that this structure has peaks in the range of 26-31° and in the range of 40-44°. However, CaMgSi2O6 does not have the characteristic intense peak near 28.8°, which is characteristic of our glasses and the above SiO2 structures. In addition, CaMgSi2O6 has intense peaks in the region of 36-36°, and for the glasses obtained in our work there are no corresponding manifestations in the diffraction patterns. The significant blurring of the peak with a maximum near 28.8° indicates that the formation of silicate structures (if they occur) is insignificant. Thus, we came to the conclusion that our materials are highly disordered structures based on SiO2.

The formation of phases in glasses is described in the literature. For example, in [Ueno, S.; Tada, T.; Suzuki, Y.; Nozawa, J.; Jang, B.-K.; Sekino, T.; Crystallization of glass with Y2Si2O7-mullite eutectic composition, Ceram. Soc. Jap. 2016, 42, 13601-13604, http://dx.doi.org/10.2109/jcersj2.16044] the formation of the Y2Si2O7 phase in Y2O3-Al2O3-SiO2 glasses is shown. The powder X-ray diffraction pattern of the glass with Y2Si2O7-mullite eutectic composition showed two halos at about 2θ = 27 and 45°. The sample was also transparent and colorless, without pronounced diffraction peaks.

The text of the article has been changed:

Section 3.1, 1 Paragraph, The sentences: “X-ray diffraction analysis was carried out to study the structure of as-melted glasses. As can be seen from Fig. 1, X-ray diffraction patterns of the SMCS glasses samples doped with Pr3+ ions do not show any diffraction peaks, which indicates the amorphous nature of the glasses. A photo of polished samples of the SCMS glass doped with 1.0% Pr3+ ions is shown in Fig. 1b.” was replaced by: “X-ray diffraction analysis was carried out to study the structure of as-melted glasses. In the diffraction pattern (Fig. 1a) of the SCMS:Pr3+ glass samples, only two very weak and diffuse peaks with wide halos can be identified (in the range of 2θ = 23-35° and 39-47°), confirming the amorphous nature of the sample. This in turn makes their interpretation very difficult. Furthermore, the presence of nanocrystalline (several nm in size) inclusions of silicate phases, which may have characteristic peaks in the region of 2θ = 23-35° and 39-47°, cannot be completely excluded. The most pronounced peak around 2θ=28.8° is most likely an amorphous halo, which is characteristic for amorphous or highly disordered SiO2-based structures.

An intense peak at 28.81°, as well as less intense peaks in the region 2θ=25.99-26.12° and 28.80-29.18° are characteristic of the SiO2 crystal structure (ICDS code 96-153-2514). The position of the halo with a maximum near 28.8° is also close to the positions of the peaks of the SiO2 structure (coesite, ICDS code 96-900-0805), however, high pressure is required for the formation of this structure. Silicates also have peaks in the region of 30-47°.

The structure of CaMgSi2O6 (ICSD code 30522) has peaks in the range of 26-31° and in the range of 40-44°. However, CaMgSi2O6 does not have the characteristic intense peak near 28.8°, which is characteristic for our glasses and the SiO2 structures.

The significant blurring of the peak (Fig. 1a) with a maximum near 28.8° indicates that the formation of silicate structures (if they occur) is insignificant. In relation, we came to the conclusion that our materials are highly disordered structures based on SiO2.

Remark 2. The figures 2, 8, 9 and 11 should be enlarged and presented at a higher resolution for better data clarity. Also, in figure 11c, the transitions must be specified.

Reply:

All symbols in Figures 2, 8, 9 and 11 have been enlarged. The transitions in figure 11c is specified. All drawings with a resolution of 300 dpi. Figures 2,8,9,11 were replaced with these corrected ones.

Remark 3. At page 7 to be replaced the words: assumed in ...is assumed in the next… (line 213); A capital letter and display in...in [22] Authors display that… (line 215), documented in ...spectrum was documented between.. (line 220) with more appropriate words. Suggestion: assumed with discussed, etc.

Reply:

Thanks for your valuable comment, The following corrections have been made to the text:

Line 213: The words “…… is assumed in …..” was replaced by “….. is explained in …..”.

Line 215: the word “Authors” was replaced by “authors”.

Line 220: the words “spectrum was documented between” was replaced by “spectrum was between”.

Remark 4. At page 8, for a unified expression of the data in the paper with those in the literature, the band energies presented in lines 263-265 should also be expressed in nm.

Reply:

The text of the article has been changed:

eV(s) were converted to nanometers.

Lines 263-265: the phrase “ ….a maximum at 3.1 eV (strong) and 4.2 eV (very weak) with excitation at 5.17 eV…” was replaced by “…... a maximum at 3.1 eV (400 nm) (strong) and 4.2 eV (295 nm) (very weak) with excitation at 5.17 eV (240 nm).”

Remark 5. At page 12, line 359 (figure caption) to delete the I in (I527/I486).

Reply:

The text of the article has been changed:

Page 12, line 359 (figure caption): the symbol “I” was deleted.

Figure captures 12: “LIR (I527/I486)” was replaced by “LIR (527/486).”.

Remark 6. At page 13, the authors to give a brief explanation of the values obtained for the transfer energy and their correlation with the previously presented data.

Reply:

A brief explanation of the obtained energy transfer parameters has been included and compared to the recently documented Pr-doped optical materials. In respect to that, the additional References 33, 34 and 35 have been inscribed.

Moreover, the adequate text was added i.e.: “The estimated critical distance R0 is higher than 5 Å, hence the applied IH model validates the contribution of multipolar Pr-Pr interactions in the material under study [33]. For the comparison, the energy transfer parameters Cda=6.39 x10-43 cm8s-1 and Wda=13×106 s-1 were estimated for Pr-doped multi-component silicate photonic films [34]. In relation to our glass, this praseodymium highly-doped silicate optical system is characterized by significant D-A energy transfer rate and especially dipole-quadrupole mechanisms leading to significant quenching phenomena. Contrary, for higher Pr2O3 concentration in P2O5-Na2O -10Al2O3-10Gd2O3 glass [35], particularly the reliable fitting is documented for S=6 and dipol-dipol interaction between optically active ions takes place.”

Reviewer 2 Report

Comments and Suggestions for Authors

 Authors have prepared multicomponent silicate SrO-CaO-MgO-SiO2 glasses doped with praseodymium with three dopant percentages and examined applying various optical spectroscopy techniques. From the results, we can understand that these samples show excellent optical properties needed for upconversion, and as well as they could be used as lasing and nonlinear optical materials. The samples were well characterized in a broad range of temperatures. However, due to errors in technical and writing parts and also in the discussion part of the manuscript, in the present form, I cannot accept publishing in materials. My concerns are listed below.

1.     Even though the abstract looks big still it does not reflect the novelty of the work. The authors have to rewrite it.

2.     The introduction looks good and the authors focused mainly on upconversion via two-photon absorption and excited state absorption. Hence they should consider a few references on nonlinear absorption with a small discussion so that readers can understand the main essence of the work very well. Here I provide some of the references for authors:  Rao, A.S., 2022. Saturation effects in nonlinear absorption, refraction, and frequency conversion: A review. Optik, 267, p.169638., He, G.S., Tan, L.S., Zheng, Q. and Prasad, P.N., 2008. Multiphoton absorbing materials: molecular designs, characterizations, and applications. Chemical reviews, 108(4), pp.1245-1330.

3.     Authors can provide the necessary information for using particular three dopant percentages: 0.5, 1, and 2.5 mol.% of praseodymium.

4.     In the characterization, why did the authors use each time different dopant percentages instead of using single dopant percentages? It makes it difficult to understand the nature of the material.

5.     Check the first three sentences of section 3.1.

6.     Figures are too small and it is very difficult to visualize and understand. There is a lot of free space between the subplots. Authors should improve their plots quality mainly Figure numbers 2, 7, 8, 9, and 11.

7.     Check the caption of Fig. 2, 4 and 5.

8.     In Fig. 6, in the x- and y-axes labels authors should clearly write emission intensity and excitation power.

9.     In lifetime measurements, generally the experimental data fit with multiple exponentials instead of single. So is it the authors provided lifetime is the average of multiple lifetimes estimated from the experimental data or they observed only a single lifetime? The sample is glass so I think the data fit with at least two exponential functions.

10.  There are lot of technical errors present in the manuscript especially in upconversion. The authors should rectify this in the revision.

Comments on the Quality of English Language

The manuscript has multiple grammatical errors and mistakes in sentence formations. In the present form, it is difficult to understand.

Author Response

Thank you very much for your comments. We addressed them all carefully, which is included below.

Remark 1. Even though the abstract looks big still it does not reflect the novelty of the work. The authors have to rewrite it.

Reply:

With respect to the meaningful Reviewer comment, the abstract has been properly rewritten to reflect the work novelty.

Remark 2. The introduction looks good and the authors focused mainly on upconversion via two-photon absorption and excited state absorption. Hence they should consider a few references on nonlinear absorption with a small discussion so that readers can understand the main essence of the work very well. Here I provide some of the references for authors:  Rao, A.S., 2022. Saturation effects in nonlinear absorption, refraction, and frequency conversion: A review. Optik, 267, p.169638., He, G.S., Tan, L.S., Zheng, Q. and Prasad, P.N., 2008. Multiphoton absorbing materials: molecular designs, characterizations, and applications. Chemical reviews, 108(4), pp.1245-1330.

Reply:

Thank you for submitting interesting reviews. We have added a short description to the article:

Introduction, after the first paragraph was added the text:

“In [4], a detailed analysis of saturation effects that occurred in various fields of nonlinear optics and considered the nonlinear optical properties of various optical materials with fast nonlinear optical response, which can be promising candidates for photonic applications, such as optical communications, optical limiters, optical data storage, information processing, passive laser mode locking, etc. In [5], several multiphoton active materials and major applications of multiphoton excitation were described, including and pumped lasing to achieve tunable up-conversion of coherent light.”

The provided references have been added, with respect to that all references starting from [4] were renumbered:

[4] Rao, A.S.; Saturation effects in nonlinear absorption, refraction, and frequency conversion: A review, Optik, 2022, 267, 169638, https://doi.org/10.1016/j.ijleo.2022.169638

[5] He, G.S., Tan, L.S., Zheng, Q. and Prasad, P.N.; Multiphoton absorbing materials: molecular designs, characterizations, and applications. Chemical reviews, 2008, 108, 1245-1330, https://doi.org/10.1021/cr050054x

References numbers were changed:

[4] to [6], [5] to [7], [6] to [8], [7] to [9], [8] to [10], [9] to [11], [10] to [12], [11] to [13], [12] to [14], [13] to [15], [14] to [16], [15] to [17], [16] to [18], [17] to [19], [18] to [20], [19] to [21], [20] to [22], [21] to [23], [22] to [23], [23] to [25], [24] to [26], [25] to [27], [26] to [28], [27] to [29], [28] to [30], [29] to [31], [30] to [32].

Remark 3.     Authors can provide the necessary information for using particular three dopant percentages: 0.5, 1, and 2.5 mol.% of praseodymium.

Reply:

Praseodymium concentrations of 0.5, 1, and 2.5% were chosen to maintain the efficient absorption qualities of the optically active ions especially at excitation wavelength 445 nm. Furthermore, for this Pr3+ content, the interionic processes are still quite ineffective to suppress the involved luminescence within examined spectral regions.

The text of the article has been changed:

Materials and Methods, P.3, 1 Paragraph, line 104:

after the words…..1.0; 2.5 at.% Pr3+, respectively).was added the sentence: “The choice of active Pr3+ ion concentrations in SCMS glasses is determined by a tradeoff between the praseodymium absorption qualities especially at excitation wavelength 445 nm and the ability to measure the up-conversion spectra of the material in the ultraviolet region as well as an emission in a wide spectral range.

Remark 4.     In the characterization, why did the authors use each time different dopant percentages instead of using single dopant percentages? It makes it difficult to understand the nature of the material.

Reply:

According to the meaningful Reviewer remark, the concentration of the Pr3+ has been revised for the presented examinations. Particularly, the results have been shown for the glass samples revealing their effective qualities for some types of experiments. Exemplary, a concentration of 2.5% Pr3+ was selected to display the absorption spectra (more details can be perceived) while up-converted and down-converted luminescence spectra are mainly effective for 1% Pr3+.

Remark 5. Check the first three sentences of section 3.1.

Reply:

Thank you for your comment. We have revised the description of the results of the diffraction analysis in our glasses. The answer to your question is provided above in the response to the first reviewer to question 1.

We have completely replaced the description of Fig. 1 and made appropriate changes to the text of the article. New version of Fig. 1 description is presented above in the responses to the first reviewer to question 1.

Remark 6. Figures are too small and it is very difficult to visualize and understand. There is a lot of free space between the subplots. Authors should improve their plots quality mainly Figure numbers 2, 7, 8, 9, and 11.

Reply:

Figure numbers 2,7,8,9,11 were replaced with corrected ones. All drawings with a resolution of 300 dpi.

Remark 7. Check the caption of Fig. 2, 4 and 5.

Reply:

The captions for the figures have been corrected.

Figure 2. The captures “Absorption spectra SCMS glasses doped with 0.5 (1), 1.0 (2), and 2.5% (3) Pr3+ ions; inset shows Pr3+ concentration dependence on the absorption coefficient of 3H43P0 transition (а); effect of temperature on SCMS:2.5% Pr3+ absorption bands (b).

was replaced by: “Absorption spectra of SCMS glasses doped with 0.5 (1), 1.0 (2), and 2.5% (3) Pr3+ ions (inset shows the dependence of the absorption coefficient of 3H43P0 transition from Pr3+ concentration); effect of the temperature on SCMS:2.5% Pr3+ absorption bands (b)”.

Figure 4. The capture “The refractive indices of the SCMS:2.5% Pr3+ glass as a function of the wavelengths” was replaced by: “The dependence of refractive indices of the SCMS:2.5% Pr3+ glass from the wavelengths”.

Figure 5. The capture “The up-converted UVC luminescence of SCMS:1%Pr3+ glass excited at 445 nm” was replaced by “The up-converted UV luminescence of SCMS:1%Pr3+ glass excited at 445 nm”.

Remark 8. In Fig. 6, in the x- and y-axes labels authors should clearly write emission intensity and excitation power.

Reply:

The text of the article has been changed:

Labels on the axes of Fig. 6 is more pronounced.

Fig. 6 was replaced with a corrected one.

Remark 9. In lifetime measurements, generally the experimental data fit with multiple exponentials instead of single. So is it the authors provided lifetime is the average of multiple lifetimes estimated from the experimental data or they observed only a single lifetime? The sample is glass so I think the data fit with at least two exponential functions.

Reply:

We are grateful for this reliable Reviewer remark, the single exponential equation could be used only for luminescent decay curves related for the samples containing a considerably low praseodymium concentration. Because, we investigate the glass samples doped with higher luminescent ions content, two-exponential functions were employed to estimate the average experimental lifetimes of the involved excited states.

Remark 10. There are lot of technical errors present in the manuscript especially in upconversion. The authors should rectify this in the revision.

Reply:

The technical errors especially in upconversion have been revised.